# How to Improve Health with Biological Agents—Narrative Review

**DOI:** 10.3390/nu14091700

**Published:** 2022-04-20

**Authors:** Anna Zawistowska-Rojek, Stefan Tyski

**Affiliations:** 1Department of Antibiotics and Microbiology, National Medicines Institute, Chelmska 30/34, 00-725 Warsaw, Poland; s.tyski@nil.gov.pl; 2Department of Pharmaceutical Microbiology, Medical University of Warsaw, Banacha 1b, 02-097 Warsaw, Poland

**Keywords:** probiotic, prebiotic, synbiotic, postbiotic, paraprobiotic, psychobiotic

## Abstract

The proper functioning of the human organism is dependent on a number of factors. The health condition of the organism can be often enhanced through appropriate supplementation, as well as the application of certain biological agents. Probiotics, i.e., live microorganisms that exert a beneficial effect on the health of the host when administered in adequate amounts, are often used in commonly available dietary supplements or functional foods, such as yoghurts. Specific strains of microorganisms, administered in appropriate amounts, may find application in the treatment of conditions such as various types of diarrhoea (viral, antibiotic-related, caused by *Clostridioides difficile*), irritable bowel syndrome, ulcerative colitis, Crohn’s disease, or allergic disorders. In contrast, live microorganisms capable of exerting influence on the nervous system and mental health through interactions with the gut microbiome are referred to as psychobiotics. Live microbes are often used in combination with prebiotics to form synbiotics, which stimulate growth and/or activate the metabolism of the healthy gut microbiome. Prebiotics may serve as a substrate for the growth of probiotic strains or fermentation processes. Compared to prebiotic substances, probiotic microorganisms are more tolerant of environmental conditions, such as oxygenation, pH, or temperature in a given organism. It is also worth emphasizing that the health of the host may be influenced not only by live microorganisms, but also by their metabolites or cell components, which are referred to as postbiotics and paraprobiotics. This work presents the mechanisms of action employed by probiotics, prebiotics, synbiotics, postbiotics, paraprobiotics, and psychobiotics, together with the results of studies confirming their effectiveness and impact on consumer health.

## 1. Introduction

The human gut microbiome consists of over 1000 species of bacteria [1,2]. The composition of this microbiome varies depending on age. In infants, it is determined by several factors, including the method of delivery (natural birth, caesarean section), the method of feeding (breastfeeding, modified milk), the use of antibiotics, as well as environmental factors. The predominant microbial group in the gut microbiome of breastfed newborns is the bacteria of the genus *Bifidobacterium*. On the other hand, in children born by caesarean section, a lower number of *Bifidobacterium* and *Bacteroides* bacteria are observed, while the number of opportunistic microorganisms, such as *Enterococcus*, *Enterobacter*, *Clostridium*, or *Klebsiella*, increases [3]. In healthy adults, the following types are dominant: *Firmicutes*, *Bacteroidetes*, *Actinobacteria*, *Proteobacteria*, and *Verrucomicrobia* [4]. Intestinal bacteria support the digestive system, synthesise vitamins, are responsible for stimulating the immune system, communicating with the intestinal epithelium, and may also influence the host’s behaviour [1]. In addition, they break down carbohydrates and fatty acids that are difficult to digest, producing short-chain fatty acids (SCFA) [4]. A lack of balance in the composition of the gut microbiome may cause numerous diseases, such as allergies, type I diabetes, inflammatory bowel disease, necrotizing enterocolitis, or obesity [3].

Some microorganisms exhibit a positive influence on the health of the host. A group of microorganisms that are safe for humans (GRAS status—Generally Recognized as Safe) and, above all, live microorganisms which, when administered in sufficient numbers, bring health benefits to the host, are referred to as probiotics [5,6,7]. In addition to the term “probiotics”, terms such as prebiotics, synbiotics, paraprobiotics, postbiotics, or psychobiotics can also be found (Table 1).

## 2. Probiotics

Probiotic microorganisms include bacteria belonging to different species as well as yeast. Bacteria belonging to the genera *Lactobacillus* and *Bifidobacterium* are the most commonly used microorganisms in various probiotic products (yoghurt, dietary supplements, or medicines). In addition, probiotic products may also contain bacteria belonging to species such as: *Streptococcus thermophilus*, or *Lactococcus lactis,* and the genera *Bacillus*, *Enterococcus*, as well as yeast—*Saccharomyces boulardii* (Table 2). Some studies have reported that live microalgae (e.g., *Chlorella* sp., *Arthrospira* sp., *Schizochytrium* sp.) can also be used as probiotics, especially in marine cultures, where they are expected to improve health and survival of marine animals [16]. Despite the proven positive impact on human and animal health of compounds extracted from algae (prebiotic effect), the evidence for probiotic benefits is still insufficient [16,17].

The bacteria belonging to the genera *Lactobacillus* and *Bifidobacterium* are Gram-positive lactic acid bacteria (LAB). They occur naturally in the digestive tract of both humans and animals. Some of the strains belonging to the genera listed are characterised by probiotic properties. When administered in sufficient numbers, they may exert a beneficial health effect on the host [5,7]. The criteria to be met by a strain classified as a probiotic organism are laid down by the FAO (Food and Agriculture Organization of the United Nations) and WHO (World Health Organization). The most important of these criteria include the origin from the human microbiome [5,6], and the absence of pathogenicity, i.e., the microorganisms should have GRAS status. According to the ISAPP guidelines (International Scientific Association for Probiotics and Prebiotics), probiotics must have strictly defined affiliation to the strain level [18,19,20]. To achieve health benefits, it is necessary to apply the minimum number of bacteria or yeast cells (colony forming units—CFU) in a daily dose, ranging from 10^8^–10^11^ CFU [21].

Probiotic microorganisms display both local and systemic impact on the host’s organism. The principal benefits from the consumption of probiotic microbes include stimulation of the immune system, production of antimicrobial substances, regulating the composition of the gut microbiome, and antimutagenic and anticancer effect [20,28,29,30]. In addition, probiotics are involved in the synthesis of vitamins, mainly from group B (B1, B2, B12), as well as vitamin K. They increase the bioavailability of certain elements, e.g., copper, calcium, iron, zinc, or manganese. An important function of probiotic microorganisms is also to stimulate intestinal peristalsis and to stimulate the differentiation and multiplication of gastrointestinal cells by supplying short-chain fatty acids produced during fermentation processes. Probiotics also reduce the absorption of exogenous cholesterol through its conversion to coprostanol, and the level of procarcinogens and carcinogens in the lumen of the gastrointestinal tract as a result of inhibiting the development of the gut microbiome, the metabolism of which is associated with the production of nitroso compounds. Probiotics also seal the intestinal mucosal barrier through the synthesis of cytoprotective substances, which may protect against the bacterial translocation phenomenon [31,32,33].

Probiotic bacteria exhibit antagonistic action against numerous bacterial pathogens of the gastrointestinal tract, such as *Salmonella enterica*, *Shigella sonnei*, enteropathogenic strains of *Escherichia coli*, *Staphylococcus aureus*, *Campylobacter jejuni,* or *Clostridioides difficile*. They impede the adhesion of these pathogens to the intestinal mucosa through competition for receptors, and they also inhibit their proliferation through competition for nutrients and by producing substances with antimicrobial activity, such as bacteriocins, organic acids, and hydrogen peroxide [33,34,35]. Certain lactobacilli can produce antimicrobial peptides, known as bacteriocins, which prevent the proliferation of certain pathogens. Bacteriocins are small cationic molecules consisting of approximately 30–60 amino acids. Bacteriocins are divided into four main types based on their basic structures, molecular weights, post-translational modifications, and genetic features [36]. Bacteriocins exhibit bactericidal or bacteriostatic activity against susceptible microorganisms. The mechanism of action of these compounds consists in destabilising the cytoplasmic membrane of susceptible bacteria by forming poration complexes and ion channels, as a result of which the efflux of small molecules such as potassium, magnesium and phosphorus ions, amino acids, and ATP, occurs. The membrane potential and the pH gradient are disturbed, and the function of the proton pump is inhibited. Low levels of ATP and ion deficiency in the cell result in the inhibition of the synthesis of DNA, RNA, proteins, and polysaccharides [37,38,39,40]. Bacteriocins can also induce cell lysis by interacting with teichoic, lipoteichoic, or teichuronic acids, which are components of the cell wall. The autolytic enzymes that are bound to these acids are released and activated, and then the cell undergoes autolysis. Nisin is the best known bacteriocin, which is produced by the strain of *Lactococcus lactis* subsp. *lactis*, as well as the strains of the genus *Streptococcus* [39,41,42]. It is characterised by a wide range of antibacterial activity, which is targeted at both Gram-positive bacteria (*Lactococcus*, *Lactobacillus*, *Streptococcus*, *Staphylococcus*, *Micrococcus*, *Pediococcus*, *Listeria,* and *Mycobacterium*) and Gram-negative bacteria (*E. coli*, *Salmonella*), and it also inhibits the growth and formation of bacterial spores of the genus *Bacillus* and *Clostridium* [37,40].

Probiotic bacteria also produce organic acids (mainly lactic and acetic acids) and hydrogen peroxide, which also have an antibacterial effect [38]. Organic acids decrease the pH in the digestive tract, which leads to the inhibition of the biochemical activity of microorganisms through undissociated acid molecules [37]. The undissociated form of the organic acid penetrates the bacterial cell and dissociates inside the cytoplasm. A decrease in the intracellular pH or intracellular accumulation of the ionized form of organic acids may lead to the death of the pathogen [35,38]. However, hydrogen peroxide inhibits growth and kills the bacteria that are not able to produce enzymes such as catalase or peroxidase [37]. Probiotic bacteria are capable of producing the so-called deconjugated bile acids which are derivatives of bile salts. Deconjugated bile acids show stronger antimicrobial activity in comparison to the bile salts synthesized by the host’s organism. It is unclear how probiotics protect themselves against their own bactericidal metabolites or whether they are resistant to deconjugated bile acids in general [33,38].

The immunostimulatory and immunomodulatory properties of probiotics are also highly important. These microorganisms cause the stimulation of the immune system linked to gastrointestinal mucosa, primarily through the increased production of immunoglobulins (mainly of the sIgA class), increased activity of macrophages and lymphocytes, and the stimulation of γ-interferon production [30,33,43,44]. Furthermore, they participate in the restoration of a proper balance between two subpopulations of lymphocytes—Th1 and Th2 [43]. Probiotics can affect the immune system through secreted metabolites, the components of a cell wall and DNA that are recognized by specialized host cells [33]. The components of the cell wall of LAB group bacteria stimulate macrophages, which, through the production of oxygen free radicals and lysosomal enzymes, may destroy microbial cells. Probiotics can also stimulate the production of cytokines in immunocompetent cells of the digestive tract [33].

Probiotics also exhibit anti-cancer activity by modifying the composition of the gut microbiome, the production of compounds with anti-cancer activity, such as i.e., short- chain fatty acids, inhibition of proliferation and induction of apoptosis in cells, improvement of the impermeability of the intestinal barrier, as well as the enhancement of the host’s immunity by secreting anti-inflammatory molecules [45,46,47]. Moreover, LAB can restrict the growth of the bacteria that synthesize enzymes, such as, e.g., β-glucosidases, β-glucuronidases, azoreductases, which catalyse the transformation of pro-carcinogenic to carcinogenic compounds, and also remove carcinogenic compounds from the diet or created by pathogenic bacteria in intestines, by shortening intestinal transit time [37]. Probiotic bacteria can inhibit the activity of nitroreductase (which is responsible for the synthesis of nitrosamines) as well as other mutagenic substances, such as nitrogen dyes or mycotoxins [30,37,46].

One of the features that should characterise a probiotic strain is its ability to adhere to the cells of mucosal epithelium and cell lines, due to which these microorganisms are able to reduce the adhesion of pathogenic microorganisms to the host’s cell surface [6]. The adhesion of microorganisms to the surface of intestinal cells allows colonization to be extended, which is important for the modulation of the immune response and may also influence the ongoing repair processes in the damaged intestinal mucosa [48,49]. Such properties belong to the mechanisms protecting the host’s organism against the colonization by pathogenic microorganisms; they promote the activity of metabolites produced by probiotics (e.g., SCFA) and also influence immunomodulatory activity [49,50]. On the other hand, the ability of probiotic bacteria to co-aggregate is one of the mechanisms that hinder the colonization of the intestine by pathogenic bacteria [33].

Probiotics display a well-documented activity in the prevention of health problems, including digestive disorders such as constipation, infection-induced diarrhoea, antibiotic-induced diarrhoea, irritable bowel syndrome, diarrhoea caused by *Clostridioides difficile* in adults and children, ulcerative colitis, Crohn’s disease, colorectal cancer, as well as allergic disorders such as atopic dermatitis (eczema) or allergic rhinitis [20,36,38,42,51] (Table 3). Phase 3 of clinical research has proved the effectiveness of probiotics containing *Lactobacillus acidophilus* CL1285 and *Lactobacillus casei* LBC80R strains in preventing and shortening the duration of antibiotic-induced diarrhoea and infection with *C. difficile* [52].

However, it should be noted that probiotics are live microorganisms, and despite having GRAS status and being considered safe for the consumer may cause side effects such as systemic infections (e.g., sepsis), excessive immunological stimulation–especially in immunocompromised people and new-borns, harmful metabolic effects, or transfer of genes (e.g., those encoding antibiotic resistance) [53].

## 3. Prebiotics

In 2017, ISAPP defined prebiotics as substrates that are selectively used by the host’s microorganisms and provide health benefits [8]. In order for a compound to be classified as a prebiotic it must meet the following criteria: resistance to low gastric pH, no hydrolysis by mammalian enzymes and no absorption in the digestive tract. Prebiotic substances should also be fermented by intestinal microorganisms for which they provide growth stimulation [67,68].

Due to the fact that prebiotics are not digested (or only partially digested) in the upper part of digestive tract, they are able to reach the large intestine where they are selectively fermented by microorganisms. This fermentation may influence the increase in the expression or alterations in the composition of short-chain fatty acids, the growth of faecal mass, the reduction in large intestine’s pH, the reduction in the amount of nitrogen end products and faecal enzymes, as well as the improvement in the functioning of the immune system, which has a beneficial impact on the host’s health. Furthermore, a prebiotic must endure production conditions so that it is not damaged, degraded, or chemically altered during the process, and it must remain available for the metabolism of intestinal bacteria [33,67]. The most common prebiotics include saccharides, which, depending on the number of combined simple sugars, are classified as: disaccharides, oligosaccharides, and polysaccharides (Figure 1). The best confirmed health-promoting properties are possessed by oligosaccharides such as: fructooligosaccharides (FOS), galactooligosaccharides (GOS), isomaltooligosaccharides (IMO), transgalactooligosaccharides (TOS), xylooligosaccharides (XOS), soybean oligosaccharides (SBOS), and mannanoligosaccharides (MOS). Polysaccharides–such as inulin, starch, cellulose, hemicellulose, or pectins– and disaccharide-lactulose [8,33,67,68,69] are also prebiotics.

Prebiotics stimulate the growth and activity of lactic acid bacteria in the human digestive tract. The products of saccharide metabolism are short-chain fatty acids, such as butyric acid, acetic acid, or propionic acid, which the host organism can use as an energy source [44,69]. Moreover, prebiotics modulate lipid metabolism, increase calcium absorption, have a positive effect on the immune system, and reduce the risk of diseases affecting the large intestine-cancers, Crohn’s disease, or irritable bowel syndrome [69].

Lactulose, being a combination of galactose and fructose [70], is considered an ideal prebiotic; it can limit the growth of intestinal bacteria, including *Clostridium*, *Bacteroides*, *Enterobacterales*, and also enhance the growth of *Bifidobacterium*, *Lactobacillus,* and *Streptococcus* [67,69]. In vitro studies have demonstrated that the lowering of the pH of faecal samples due to the presence of fatty acids and lactates resulting from lactulose ingestion reduces the growth of pathogenic *C. difficile* and *Bacteroides* spp. [69,71].

Fructooligosaccharides (FOS) occur naturally, e.g., in onions, garlic, tomatoes, rice, wheat, rye, Jerusalem artichoke, nectarines, papaya, or banana peels [72]. These compounds ferment to lactates and short-chain fatty acids, lower the pH, produce gas in the intestines, and increase the bioavailability of important elements, such as calcium, manganese, iron, and zinc [71,73]. Moreover, they stimulate the growth of the bacteria of the genus *Bifidobacterium* and exhibit a strong influence on the intestinal mucosa, with particular emphasis on their role in inflammatory bowel diseases [72].

Galactooligosaccharides (GOS) are composed of 2–5 galactose monomers attached to glucose [74]. These compounds participate in the modulation of the colon microbiome, stimulate the growth of *Lactobacillus* and *Bifidobacterium*, while inhibiting the growth of *Clostridium*, *Bacteroides,* and enterobacteria. In addition, GOS inhibit the adhesion of pathogenic bacteria to the intestinal epithelium, lowering cholesterol levels and blood pressure, and boosting immunity [74,75]. These compounds are often used as sweeteners. They have a protective effect on probiotic strains during production processes, such as lyophilisation, resulting in the formation of a symbiotic preparation [75]. Due to the high stability of GOS in high temperature and acidic environment, they are used as additives in baby food, dietary supplements, sauces, soups, ice cream, beverages, bread, animal feed, etc. [71].

Inulin is a naturally occurring carbohydrate commonly found in leeks, onions, wheat, asparagus, garlic, Jerusalem artichoke, and chicory [71]. Inulin is used to treat irritable bowel disease and colon cancer. Moreover, it stimulates the growth of beneficial bacteria and also reduces many factors of intestinal disorders [72]. Jackson et al. [76] found that daily consumption of 10 g of inulin significantly lowers insulin levels in the studied group of women and men. A trend in the reduction in triacylglycerol levels has also been observed [76].

## 4. Synbiotics

The activity of probiotics may be enhanced, as well as supplemented by prebiotics [33]. In 1995, Gibson and Roberfroid [77] introduced the term synbiotics as a combination of probiotics and prebiotics having a synergistic effect, which consists of introducing into the gastrointestinal tract an appropriate ingredient which stimulates the growth and/or metabolism of the normal gut microbiome, which is conducive to improving the health of the host. In 2019, ISAPP [9] defined synbiotics as a mixture consisting of living microorganisms and a substrate selectively used by host microorganisms which provide health benefits to the host. Such a combination of a probiotic and prebiotic should be properly tested and have a proven synergistic effect in comparison with placebo [9].

Some authors [9,78] distinguish two types of synbiotics: a complementary and a synergistic one. A complementary synbiotic consists of a probiotic and a prebiotic that work independently to achieve one or more health benefits. Both components of this type of synbiotic must meet the minimum criteria specified for probiotics and prebiotics. A synergistic synbiotic, on the other hand, is designed in such a way that the substrate is intended for selective use by microorganisms. The microorganism is to provide health benefits to the host, and the appropriately selected substrate is to stimulate the growth or activity of selected microorganisms. Importantly, none of the components of such a synbiotic has to meet the minimum criteria specified for pro- and prebiotics [9,78]. The synbiotics are most often composed of the bacteria of the genera *Lactobacillus*, *Bifidobacterium,* and *Streptococcus* as a probiotic component, and oligosaccharides, inulin, or fibre as a prebiotic component [9,78]. Due to the application of this type of combination, the survivability of probiotic microorganisms in the gastrointestinal tract is improved. Prebiotics, in the right combination, can serve as a substrate for the growth of probiotic strains or fermentation processes. Thanks to prebiotic substances, probiotic microorganisms are more tolerant of environmental conditions such as oxygenation, pH, or temperature prevailing in a given organism [33]. In order for probiotics to reach their destination after oral supplementation, they must survive in the acidic environment of the stomach, and only after reaching the intestines is it possible to colonize them. Because of that, microencapsulation is a frequently applied process, which, apart from probiotic microorganisms, may also affect enzymes, natural bioactive substances, prebiotics, or gaseous materials [79]. Microencapsulation protects against the harsh and changing conditions of the gastrointestinal tract and promotes the release of certain substances, usually in the colon. Microcapsules also protect the contained load during the stabilization process, and during storage in a wide range of temperatures, they can significantly extend the shelf life of a given product. A common form is microencapsulated dietary fibre, e.g., inulin combined with a probiotic microorganism. Such a combination increases the stability of the probiotic storage, protection during the course of processing, and also protects the microorganisms during the passage through the gastrointestinal tract. The use of inulin–a thermally stable compound–for the construction of microcapsules is intended to protect the probiotic; moreover, it also fulfils its functions as a prebiotic [79]. In the research conducted by dos Santos et al. [80] it was demonstrated that 10% inulin applied as a coating agent protects the *L. acidophilus* La-5 strain during the spray-drying process, and also protects the tested strain under conditions with the addition of artificial gastrointestinal juices. Similar studies were conducted by Atia et al. [81], in which they demonstrated the capability of inulin, being an additive to alginate microcapsules, to protect the probiotic strains of *Pediococcus acidilactici* UL5, *L. reuteri,* and *L. salivarius*.

The abundant evidence indicates a synergistic and complementary effect of pro- and prebiotics against intestinal microorganisms. The conducted studies have demonstrated that the use of synbiotics may modulate metabolic activity in the intestine, influence the ratio of *Firmicutes* type bacteria to *Bacteroidetes* bacteria, inhibit the growth of pathogenic microorganisms through direct antagonism or competitive exclusion, e.g., of the strains of the genus *Klebsiella*, as well as *E. coli* or *C. difficile*, in addition to accelerating the regeneration of the gut microbiome [33,79]. Synbiotics reduce the concentration of undesirable metabolites as well as inactivate nitrosamines and carcinogens. Their use leads to a significant increase in the level of short-chain acids, ketones, carbon disulphide, and methyl acetate, which potentially has a positive effect on the host’s health [82]. The properties of synbiotics include anti-cancer and anti-allergic activity. They also counteract rotting processes in the intestine and prevent constipation and diarrhoea [83], and also find application in the treatment of ulcerative colitis [84]. It is worth noting that the positive effect of synbiotics depends on the appropriate combination of probiotics and prebiotics, as well as their dose [79].

The application of synbiotics displays promising activity in two main categories of human diseases: inflammation-related and metabolic diseases [85]. A highly important component of healthy intestines that is associated with diseases belonging to both categories is the production of short-chain fatty acids (acetate, propionate, butyrate) through bacterial fermentation of dietary fibre in the colon [72,85]. Butyrate plays a very important role in modulating enteritis; its deficiency is often associated with inflammatory bowel disease (IBD) [85]. Irritable bowel syndrome is, in turn, a disease of the gastrointestinal tract that has no strictly defined causes, manifested mainly by abdominal pain, as well as alternating constipation and diarrhoea [86]. The conducted clinical trials have demonstrated promising results in the alleviation of IBS symptoms after the use of synbiotics. The research conducted by Lee et al. [87] with the use of a synbiotic containing the strains *L. rhamnosus*, *L. acidophilus*, *L. casei*, *L. bulgaricus*, *L. plantarum*, *L. salivarius*, *B. bifidum,* and *B. longum*, as well as FOS, inulin, elm bark, and herb bennet, demonstrated a reduction in intestinal symptoms (abdominal discomfort, abdominal distension, frequency of forming stools) and fatigue compared to the placebo group. In the research by Min et al. [88], a reduction in general disease symptoms was observed in patients with IBS compared to the control group after treatment with a synbiotic in the form of yoghurt containing high doses of the strains: *B. animalis* subsp. *lactis* BB-12 (10^11^ CFU/dose), *S.*
*thermophilus* (3 × 10^9^ CFU/dose), and *L. acidophilus* (10^9^ CFU/dose), and of acacia fibre. In turn, in the research by Šmid et al. [89], in which patients were administered 180 g of a synbiotic (fermented milk) containing *L*. *acidophilus* La-5 (1.8 × 10^7^ CFU/g), *Bifidobacterium* BB-12 (2.5 × 10^7^ CFU/g), *S. thermophilus,* and dietary fibre, no beneficial effects were observed compared with the control group. Further clinical trials with the use of synbiotics are necessary to obtain a more definitive opinion about their influence on IBS [86].

Synbiotics also have a potential anti-carcinogenic effect, but the obtained experimental results are inconclusive and are the subject of investigation and discussion. Synbiotics may facilitate the death of a damaged cell in the colon, may enhance the colonization of the intestines by microorganisms, stimulate the growth and activity of probiotics in the presence of prebiotics, may increase SCFA production, and also display immunomodulatory activity in addition to improving intestinal metabolic activity [90,91,92]. In the studies conducted by Rafter et al. [93] with the use of a synbiotic containing inulin enriched in oligofructose and *L. rhamnosus* GG (LGG) as well as *B. lactis* BB-12 (BB12), changes in the faecal microbiota were observed in the control group (increase in the number of *Bifidobacterium* and *Lactobacillus*, decrease in the number of *Clostridium perfringens*) compared to the placebo group. Moreover, a decrease in the proliferation of the large intestine and an improvement in the intestinal barrier function was observed [93]. On the other hand, in the studies conducted by Flesch et al. [94], patients with colorectal cancer were administered a synbiotic consisting of the following strains: *L. acidophilus* NCFM, *L. rhamnosus* HN001, *L. paracasei* LPC-37, and *B. lactis* HN019, and fructooligosaccharides (FOS). It was observed that the perioperative administration of the above-mentioned synbiotic decreased the rates of postoperative infections. In the course of conducted research, Krebs [95], however, did not find differences in the postoperative progress and the complication rate between the groups of people taking prebiotics, synbiotics, and the placebo group.

Synbiotics may also influence the control of the lipid profile [83,96]. In the studies of Karimi et al. [97] it was observed that a 12-week dietary supplementation with synbiotics including the following strains: *L. acidophilus* 3 × 10^10^ CFU/g, *L. casei* 3 × 10^9^ CFU/g, *L. bulgaricus* 5 × 10^8^ CFU/g, *L. rhamnosus* 7 × 10^9^ CFU/g, *B. longum* 1 × 10^9^ CFU/g, *B. breve* 2 × 10^10^ CFU/g and *S. thermophilus* 3 × 10^8^ CFU/g, and inulin, was conducive to increasing the level of HDL (high density lipoprotein) and reducing LDL (low density lipoprotein) in patients with polycystic ovary syndrome. In turn, in the studies conducted on a group of pregnant women, Taghizadeh et al. [98] observed a significant reduction in the level of triacylglycerols, VLDL (very low-density lipoprotein), and glutathione, but recorded no influence of a synbiotic containing *L. sporogenes* (1 × 10^7^ CFU/g) and inulin on the overall level of cholesterol, HDL and LDL. Patients with type 2 diabetes consumed synbiotic bread containing *L. sporogenes* (1 × 10^8^ CFU/g) and inulin; after eight weeks there was a significant decrease in triacylglycerol and VLDL levels, a decrease in the ratio of total cholesterol to HDL and a significant increase in HDL levels. No impact on the level of total cholesterol and LDL was observed [99].

## 5. Postbiotics

A relatively new concept that has been appearing in the literature from approximately 10 years is the term “postbiotics”. In 2021, ISAPP defined postbiotics as preparations containing inanimate microorganisms and/or their components that induce a health benefits on the host [10]. The term “postbiotic” was introduced to distinguish live microbial cells, i.e., probiotics, from a bioactive product that contains dead microorganisms and their metabolites, such as soluble factors secreted by live bacteria or released after bacterial lysis of probiotic strains, including enzymes, peptides, bacteriocins, cell surface proteins, polysaccharides, vitamins, organic acids, SCFA, and amino acids [15,100,101]. In addition, the term “postbiotics” refers to preparations that contain detailed information on the microbial strains present in the product, matrix, and the description of the inactivation method that the microorganisms have been subjected to, since the composition of the postbiotic may depend on its type as well as the detailed composition of the final product [10]. Purified microbial metabolites and vaccines are not classified as postbiotics [10].

Postbiotics stimulate the gut microbiome and support the gut’s immune function. They can also inhibit the multiplication of pathogenic microorganisms, because this group of products includes, among others, bacteriocins, organic acids, peptides, fatty acids, and hydrogen peroxide [10,15,102]. Moreover, antioxidant, anti-carcinogenic, immunomodulatory, and anti-obesity effects have also been demonstrated [15,44,100]. Postbiotics may also influence the gut microbiome indirectly, e.g., through the quorum sensing mechanisms and “quorum quenching molecules” [10].

The antimicrobial properties of postbiotics depend on numerous factors, including the substances used in the production of bacterial strains, the substances they produce, and their concentration. The organic acids with the strongest inhibitory effect on the growth of pathogenic microorganisms include lactic acid and acetic acid, which lower the intracellular pH and integrity of the cell membrane [102]. Bacteriocins also display strong antibacterial activity, which depends on the size, mechanism of action, and spectrum of inhibiting microbial growth. Bacteriocins impact bacterial peptide structures, and inhibit the germination of spores and the formation of pores in the cell membranes of pathogens [101,102,103]. Fatty acids and their derivatives also inhibit the growth of pathogenic microorganisms. Long-chain fatty acids such as eicosapentaenoic acid (EPA) inhibit the growth of Gram-negative bacteria. Other acids, such as lauric and myristic acids, limit the growth of microorganisms. Fatty acids affect bacteria by increasing the permeability of the cell membrane, lysis of cells, disrupting the electron transport chain, disrupting the structure and activity of enzymes, and inducing morphological/functional changes in sensitive cell components, such as proteins [102]. In contrast, the mechanism of action of peptides consists in degrading membranes and inhibiting the synthesis of macromolecules [102]. The inhibitory effect of hydrogen peroxide depends on numerous factors–mainly on its concentration–and is associated with strong oxidising functions in the bacterial cell and with damage to the structure of cytoplasmic proteins [102].

A highly important feature of postbiotics is their stability, both during technological processes and storage, which is their unquestionable advantage over probiotics, in which time, temperature, and water activity have a significant influence on the stability of preparations during storage [10].

In Europe, postbiotics have been used for many years, but there is no regulation regarding this product group. European Food Safety Authority establishes regularly updated lists of the microorganisms which comply with the criteria for presumed safe use in food. This process, called the Qualified Presumption of Safety (QPS), is applicable to live microorganisms (including bacteria and yeast) used as progenitor agents for postbiotics [10]. Medicinal product-Lacteol Fort, containing in its composition inactivated strains of *Lactobacillus* LB (*Lactobacillus delbrueckii* and *Lactobacillus fermentum*) with a quantity of 1 × 10^10^ (https://amscohealthcare.com/products/lacteol-fort-sachet/; accessed on 16 December 2021) [104] is used in the treatment of chronic diarrhoea. It alleviates symptoms such as abdominal pain or flatulence and also shortens the duration of viral and bacterial diarrhoea in children [10,105]. Jeong et al. [106] administered a tyndalised strain of *Lactobacillus rhamnosus* IDCC 3201 (RHT3201) to a group of children aged 1–12 years, suffering from atopic dermatitis. A reduction in the severity of atopic dermatitis (Scoring Atopic Dermatitis—SCORAD) was observed in the postbiotic group. Moreover, a decrease in the level of eosin cationic protein and interleukin IL-31 was noted in the group of children older than 50 months [106].

Postbiotics have a considerably better safety profile compared to probiotics, as the microorganisms which are found in postbiotics have lost the ability to replicate and cannot cause bacteremia or fungemia [10], which can sometimes occur when consuming probiotics [10,53].

## 6. Paraprobiotics

A paraprobiotic is slightly similar to postbiotics and is otherwise known as “non-viable” probiotic, an inactivated probiotic which is defined as non-viable microbial cells (intact or damaged) or cell extracts which, when administered (orally or topically) in adequate amounts, benefit consumers: humans or animals [11]. Postbiotics and paraprobiotics exhibit immunomodulatory activity, which ensures health benefits for the host [11].

Paraprobiotics are isolated from microorganisms that have entirely lost their viability; the cells of these microorganisms are unable to grow in vitro [105]. Special procedures are required to obtain paraprobiotics and postbiotics from probiotic bacteria, most commonly by thermal treatment, but also by enzymatic or chemical treatment, solvent extraction, ionizing or UV radiation, high pressure, or sonication [11,15,101,103,105]. Paraprobiotics, being inactivated cells of probiotic microorganisms, contain components of probiotic cells after lysis, such as teichoic acids, peptidoglycan, and polysaccharides, e.g., exopolysaccharides, proteins associated with the cell surface, and protein fibres [101,103,107]. The method of inactivating microbial cells using various methods as well as their effect on cellular structural components and biological activity are not identical. Heat inactivation is carried out using a wide temperature range to ensure that the bacteria remaining in the suspension are killed. In in vivo tests, heat-inactivated cells exhibit potentially beneficial effects for the host at the intestinal level. Probiotics inactivated in this way were characterised by the ability to compete for the adhesion site with enteropathogens in in vitro tests carried out on the Caco-2 cell line, which may indicate a potential application, for example, in diarrhoea [101]. In addition, some of the *Lactobacillus* strains subjected to heat inactivation exhibited anti-inflammatory effects (ability to suppress inflammatory markers such as IL-5 and TNF-α, and enhance anti-inflammatory cytokines such as IL-10) and antioxidant properties (ability to remove free radicals) in in vitro and in vivo experimental models [101].

Due to their properties–such as stability in a wide range of pH and temperature or thermal treatment without loss of biochemical functionality, while not changing the sensory properties of the product–paraprobiotics seem to be of great interest for use in industry, e.g., dairy industry [11]. The preparation of functional food containing probiotics is connected with numerous challenges, such as ensuring adequate survivability, stability, and functionality of the strains used, both during the production process and during the storage of products [11,107]. The use of paraprobiotics seems to be a perfect alternative to the problems associated with the application of probiotics, particularly in the case of stability during the processes of preparation and storage of products. It is also worth considering the simplicity of their production and good distribution in food, however, these aspects require further examination [11]. Heat-inactivated probiotics are sterile and can be used in any product, regardless of its composition or product type, and the risk of contamination with live bacteria during production is relatively low [108]. Products containing paraprobiotics in their composition can be stored at room temperature; moreover, due to the absence of live microorganisms, they reduce the risk of microbial translocation and the risk of infection among consumers [107]. Therefore, they can be a safe alternative for immunocompromised individuals, such as the elderly, transplant patients, or premature babies, and may eliminate various disadvantages of probiotics [1,11,53].

Paraprobiotics are used in the treatment of, among others, diarrhoea, colitis, allergies, atopic dermatitis, and respiratory diseases; furthermore, they can regulate the immune system and the gut microbiome composition [15,105]. Sugawara et al. [109] examined the effect of a thermally inactivated *Lactobacillus gasseri* CP2305 strain on the functioning of intestines. It was observed that consuming a paraprobiotic for three weeks improved the gut microbiome environment and intestinal functions in healthy participants of the study, who were prone to constipation or frequent bowel movements. In addition, a significant increase in the number of bacteria of the genus *Bifidobacterium* and a decrease in the number of *Clostridium* cells were noted, when compared to the placebo group [109]. In contrast, a study by Nakamura et al. [110] demonstrated the effect of non-viable cells of *Lactobacillus amylovorus* CP1563 on anthropometric measurements and markers associated with lipid and glucose metabolism in people with first-degree obesity. Reductions in adipose tissue, total cholesterol, triglycerides, and LDL were observed, as well as reductions in diastolic blood pressure, blood glucose, insulin, and uric acid levels compared to the placebo group [110]. Similar conclusions were reached by Sugawara et al. [111] when administering the paraprobiotic *Lactobacillus amylovorus* CP1563 with the addition of 10-hydroxyoctadecanoic acid (10-HOA) to a group of overweight people. A significant decrease in the amount of adipose tissue was observed in patients from the test group. Moreover, changes in the composition of the gut microbiome were also recorded, the number of bacteria of the genus *Roseburia* and *Lachnospiraceae* increased, and the number of *Collinsella* bacteria decreased in the study group [111].

In addition to the terms postbiotics and paraprobiotics, some works also include the term metabiotics, which refers to low molecular weight compounds that are metabolites, signaling molecules, or fragments of dead microbial cells. Metabiotics can be used as food additives, and they can also influence the composition and functions of the gut microbiota. In addition, these compounds can also influence biochemical and behavioral responses as well as intracellular and intercellular information exchange [112,113]. Examples of compounds belonging to metabiotics are: bacteriocins, short-chain fatty acids, proteins, peptides, polysaccharides, vitamins, or nucleic acids [113].

## 7. Psychobiotics

Another group of related products are psychobiotics, defined as live microorganisms which, when taken in appropriate amounts, provide mental health benefits by interacting with the gut microbiome, enhancing cognitive functions, and modulating anxiety and stress levels [2,12,13,14,15]. Psychobiotics exhibit anti-anxiety and antidepressant effects by influencing the nervous system, and are also related to cognitive functions, memory, learning, and behaviour [13,14]. Since many probiotics release neuroactive compounds when certain conditions are met, the term psychobiotics should only be used to refer to microorganisms which have a strong, positive effect on the brain, and thus on a person’s mental health and behaviour [12]. Due to the complexity of the so-called gut-brain axis, elucidating the specific mechanisms of action of probiotic microorganisms, as well as determining how to assess the psychobiotic effects of specific strains in a probiotic product or food product, remains a challenge [14]. Sarkar et al. [114] suggests that the term psychobiotics should also be used to refer to other substances that induce beneficial changes in the microbiome, e.g., prebiotics that promote the growth of bacteria possessing psychobiotic potential. Thus, synbiotics containing bacteria with psychobiotic potential, along with prebiotics, should also be considered psychobiotics [114].

Probiotics may affect the functioning of the central nervous system in various ways. For example, they may affect it by stimulating the host cells to produce neurotransmitters (serotonin, dopamine, gamma-aminobutyric acid—GABA) [115]. In addition, some probiotic strains produce their own neurochemicals [12,116]. The strains of *L. helveticus* and *L. delbrueckii* subsp. *bugaricus* can produce norepinephrine and/or dopamine, moreover *L. brevis*, *Lact. lactis* and *L. rhamnosus* GG synthesize GABA, in turn serotonine was detected in *B. subtilis* and *L. helveticus*. Histamine, in turn, can be produced by numerous strains of both Gram-negative and Gram-positive bacteria, including by strains of *L. acidophilus* and *Bacillus* spp. [116]. On the other hand, *E. coli*, *B. cereus,* and *Lactobacillus* spp. can produce catecholamines and their precursor—2,3-dihydrophenylalanine (DOPA), which is converted to dopamine and norepinephrine [12]. Microorganisms may also alter the expression of neurotransmitter receptors in the brain and may additionally alleviate systemic inflammation by mediating an increase in anti-inflammatory cytokines and a decrease in pro-inflammatory cytokines [115]. The effects on the diversity of the gut microbiome, vagus nerve signalling through changes in tryptophan metabolism, and the production of neuroactive microbial metabolites should also be classified among the effects of probiotics [115,117]. Psychobiotics cause the level of short-chain fatty acids to increase [2,14,114,117]. The amount and type of consumed fibre greatly influence the composition of the gut microbiome and, at the same time, the amount and type of SCFAs produced. These acids regulate the host’s cellular metabolism; they influence regulating the integrity of the epithelial barrier, regulate the immune system, the inflammatory response of the organism, as well as influence the metabolism of lipids and adipose tissue. Moreover, they can influence the nervous system by increasing the integrity of the blood-brain barrier or by modulating neurotransmission [14].

Psychobiotic bacteria can stimulate an increase in production of various neurotransmitters, such as serotonin, dopamine, GABA, acetylcholine, and noradrenaline, which have the ability to directly affect the activity of the brain [14]. Serotonin is a neurotransmitter that is responsible for regulating behavioural and biological functions in the body. The lack of improper balance or improper regulation of serotonin levels may manifest itself in cardiovascular diseases, irritable bowel syndrome, or osteoporosis. Moreover, serotonin is also responsible for the regulation of cognitive functions, memory processes, and mood [14,117]. The production of serotonin is stimulated by bacteria of the genus *Enterococcus*, *Streptococcus*, and also *Escherichia* [114,117]. Dopamine, norepinephrine, and epinephrine are biogenic amines, with tyrosine being their precursor. They play an important role in motor control, learning, memory, and stress response. They also influence the cardiovascular system by regulating the metabolism of carbohydrates and fats [14]. The production of dopamine and noradrenaline is stimulated by bacteria of the genus *Bacillus*; additionally, the production of noradrenaline is also influenced by the bacteria of the genus *Escherichia* [114,117]. GABA and glutamate are responsible for the control of excitatory and inhibitory neurotransmitters. Their coordination is important for the proper functioning of processes such as neuronal excitability, synaptic plasticity, or cognitive functions, e.g., learning, memory [14,117]. GABA production is stimulated by the bacteria of both the genus *Lactobacillus* and *Bifidobacterium* [14,114,118]. Acetylcholine is the main excitatory neurotransmitter; it influences synaptic plasticity, strengthens neuronal loops and cortical dynamics during learning, and affects the excitability of neurons [14]. Its production is stimulated by strains of the *Lactobacillus* genus [14,114,117].

In the study conducted by Otaka et al. [118], patients with depression were given *Lacticaseibacillus paracasei* strain Shirota (*Lactobacillus casei* Shirota) at a dose of 8 × 10^10^ CFU/day. After 12 weeks of taking the probiotic, a significant reduction in depression symptoms was observed. This was observed together with changes in the composition of the gut microbiome–the number of *Bifidobacterium* and *Actinobacteria* bacteria in the intestine was increased [118]. Multi-strain probiotic administration containing in one capsule: *Bacillus coagulans* IS2 2 × 10^9^ CFU, *Lactobacillus rhamnosus* UBLR58 2 × 10^9^ CFU, *Bifidobacterium lactis* UBBLa70 2 × 10^9^ CFU, *Lactobacillus plantarum* UBLP40 2 × 10^9^ CFU, *Bifidobacterium breve* UBBr01 1 × 10^9^ CFU, and *Bifidobacterium infantis* UBBI01 1 × 10^9^ CFU with glutamine 250 mg, 2 times a day for 28 days, significantly reduced the level of stress in the examined students before the exam. Moreover, compared to the placebo group, a decrease in cortisol in serum, which is one of the most important stress hormones, was also observed [119]. In the research by Dickerson et al. [120], administration of probiotics (*L. rhamnosus* GG and *B. animalis* subsp. *lactis* BB-12) was associated with a reduced risk of psychiatric hospital readmission for patients with mania. The risk of re-hospitalization was 2.5–3 times lower in the study group than in the group that was receiving a placebo [120].

In the literature, one can also find the term parapsychobiotics, i.e., paraprobiotics that have a beneficial effect on mental health [15], e.g., by reducing stress [121]. However, similarly to psychobiotics, it is not clear how heat-inactivated bacterial cells can affect the gut-brain axis and alter stress responses [14,121].

Nishida et al. [122], in a study conducted on 60 healthy Japanese medical students, used a parapsychobiotic containing a heat-inactivated strain of *Lactobacillus gasseri* CP2305. It was observed that administration of this strain reduced anxiety and sleep disorders compared to the placebo group. In addition, it was demonstrated that in the test group receiving *L. gasseri* CP2305, the stress-induced decrease in *Bifidobacterium* spp. and the increase in *Streptococcus* spp. were attenuated [122].

## 8. Conclusions

Both live (probiotics) and dead microorganisms, as well as their components or metabolites (postbiotics, paraprobiotics), often in combination with prebiotic substances (synbiotics), exhibit beneficial effects on the host’s organism, confirmed in scientific research and clinical trials. Providing the organism with appropriate pro, post-, para-, pre-, and synbiotic substances exerts a positive impact on the balance of the gut microbiome and inhibits the development of pathogenic microorganisms by lowering the pH of the intestinal environment, production of short-chain fatty acids, adhesion to the intestinal mucosa cells, and competitive displacement. Biological agents–probiotics, prebiotics, postbiotics, paraprobiotics, and synbiotics–have a positive effect on digestive disorders such as constipation, diarrhoea caused by infections, post-antibiotic diarrhoea, ulcerative colitis, colon cancer, and allergic disorders, and also stimulate the immune system. Moreover, prebiotics and synbiotics also modulate lipid metabolism [10,15,20,36,38,42,51,69,90,91,92,102,105]. Psychobiotics, in turn, affect the nervous system by displaying anti-anxiety and depression-reducing effects [13,14]. It seems that the influence of the discussed biological agents on the human body is extensive and multidirectional. The continuation of the ongoing research and conducting more in-depth studies may yield interesting and important results being of relevance from the public health point of view.

## Figures and Tables

**Figure 1 nutrients-14-01700-f001:**
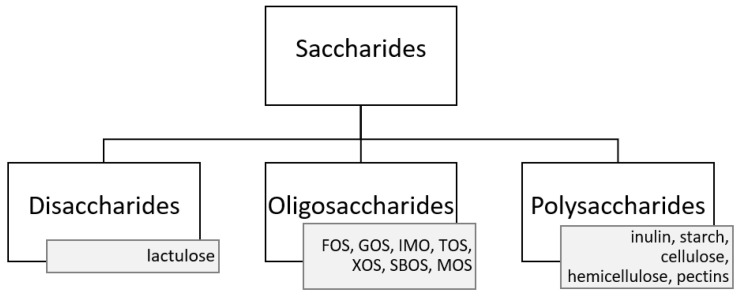
Examples of the most commonly used prebiotics [8,33,67,68,69]; FOS—fructooligosaccharides; GOS—galactooligosaccharides; IMO—isomaltooligosaccharides; TOS—transgalactooligosaccharides; XOS—xylooligosaccharides; SBOS—soybean oligosaccharides; MOS—mannanoligosaccharides.

**Table 1 nutrients-14-01700-t001:** Biological factors that can affect the health of the host.

Name	Definition	References
Probiotics	live microorganisms that, when administered in adequate amounts, confer a health benefit on the host	[7]
Prebiotics	a substrate that is selectively utilized by host microorganisms conferring a health benefit	[8]
Synbiotics	a mixture comprising of live microorganisms and substrate(s) selectively utilized by host microorganisms that confers a health benefit on the host	[9]
Postbiotics	preparation of inanimate microorganisms and/or their components that confers a health benefit on the host	[10]
Paraprobiotics	non-viable microbial cells (either intact or broken), or crude cell extracts, which, when administered (orally or topically) in adequate amounts, confer a benefit on the human or animal consumer	[11]
Psychobiotics	probiotics that confer mental health benefits to the host when consumed in a particular quantity through the interaction with commensal gut bacteria	[12,13,14,15]

**Table 2 nutrients-14-01700-t002:** Probiotic microorganisms [22,23,24,25,26].

Genus	Species
*Lactobacillus*	*L. rhamnosus* (*Lacticaseibacillus rhamnosus* *), *L. acidophilus*, *L. plantarum* (*Lactiplantibacillus plantarum* *), *L. casei* (*Lacticaseibacillus casei* *), *L. delbrueckii* subsp*. bulgaricus*, *L. brevis* (*Levilactobacillus brevis* *), *L. johnsonii*, *L. fermentum* (*Limosilactobacillus fermentum* *), *L. reuteri* (*Limosilactobacillus reuteri* *), *L. gasseri*, *L. paracasei* (*Lacticaseibacillus paracasei* *), *L. salivarius* (*Ligilactobacillus salivarius* *)
*Bifidobacterium*	*B. infantis*, *B. animalis* subsp. *lactis*, *B. bifidum*, *B. longum*, *B. breve*, *B. animalis* subsp*. animalis*, *B. adolescentis*
*Enterococcus*	*E. durans*, *E. faecium*, *E. faecalis*, *E. lactis*, *E. hirae*
*Bacillus*	*B. coagulans*, *B. subtilis*, *B. cereus*, *B. clausii*, *B. pumilus*, *B. licheniformis*
Other	*Lactococcus lactis* subsp*. lactis*, *Streptococcus thermophilus*, *Pediococcus acidilactici*, *Leuconostoc mesenteroides*, *Escherichia coli* Nissle 1917, *Saccharomyces boulardii*

* Name according to Zheng et al., 2020 [27].

**Table 3 nutrients-14-01700-t003:** Examples of the activity of probiotics confirmed in scientific research.

Disease	Probiotic Strains/Duration of Treatment	Effects of Activity	References
Rheumatoid arthritis	*L. acidophilus* 2 × 10^9^ CFU/g *L. casei* 2 × 10^9^ CFU/g *B. bifidum* 2 × 10^9^ CFU/g 8 weeks	improvement of the DAS-28 (Disease Activity Score) reduction in insulin level	[54]
Irritable bowel syndrome (IBS)	*L. acidophilus* DDS-1, 1 × 10^10^ CFU/day *B. animalis* subsp. *lactis* UABla-12, 1 × 10^10^ CFU/day 6 weeks	reduction in abdominal pain severity mitigation of IBS symptoms	[55]
	*L. casei* Zhang, 3 × 10^9^ CFU/g *B. animalis* subsp. *lactis* V9, 4 × 10^9^ CFU/g *L. plantarum* P-8, 3 × 10^9^ CFU/g 28 days	reduction in the severity of IBS symptoms, reduction in the level of interleukin-6 (IL-6) and tumour necrosis factor-α (TNF-α) reduction in the number of bacteria: *Bacteroides*, *Escherichia*, *Citrobacter*	[56]
Ulcerative colitis	*L. rhamnosus* NCIMB 30174 *L.plantarum* NCIMB 30173 *L. acidophilus* NCIMB 30175 *E. faecium* NCIMB 30176 1 × 10^10^ CFU/dose; 4 weeks	reduction in intestinal inflammation	[57]
Infant colic	*B. animalis* subsp. *lactis* BB-12^®^ (BB-12), 1 × 10^9^ CFU/day 21 days	reduction in the duration of colic in the group receiving probiotic in comparison to the group receiving placebo	[58]
Biochemical, oxidative and inflammatory markers	*L. casei* LTL 1879 1.41 ± 0.12 × 10^11^ CFU/g 3 weeks	an increase in the level of total antioxidant capacity (T-AOC) reduction in malondialdehyde (MDA) level an increase in the level of interleukin-10 (IL-10) and tumour necrosis factor-α (TNF-α) lowering the expression of *Escherichia coli*, *Enterococcus* and *Bacteroides* genes an increase in the expression of *Clostridium leptum*, *Bifidobacterium* and *Lactobacillus* genes	[59]
Chronic diarrhea	*L. plantarum* CCFM1143 3.52 × 10^9^ CFU/day 30 days	reduction in the frequency of defecation significant increase in acetic and propionic acid content change in the diversity of the gut microbiome-reduction in the quantity of *Bacteroides* bacteria	[60]
Antibiotic associated diarrhea	*L. casei* DN 114001 1 × 10^10^ CFU/dose 3 months	reduction in the incidence of diarrhoea reduction in the duration of diarrhoea	[61]
	*L. reuteri* ATCC 55730 1 × 10^8^ CFU/dose 2 × a day/28 days	reduction in the incidence of diarrhoea	[62]
Gastrointestinal symptoms	*L. johnsonii* IDCC 9203 *L. plantarum* IDCC 3501 *B. lactis* IDCC 4301 1.0 × 10^10^ CFU/capsule 8 weeks	mitigation of general symptoms (abdominal pain and flatulence) a significant increase in the level of *Lactobacillus johnsonii* and *Bifidobacterium lactis* in faeces	[63]
Atopic dermatitis	*L. plantarum* IS-10506 10^10^ CFU/day 12 weeks	lower SCORAD (Scoring Atopic Dermatitis) values in children lower levels of IL-4, IFN-γ and IL-17 IgE level did not change significantly	[64]
	*L. plantarum* CJLP133 0.5 × 10^10^ CFU/dose 2 × a day/12 weeks	lower SCORAD values in the group of children aged from 12 months to 13 years lower number of eosinophils, reduction in IFN-γ and IL-4 level	[65]
	*L. plantarum* IS-10506 2 × 10^10^ CFU/day 8 weeks	lower SCORAD score in adults lower levels of IL-4, IFN-γ and IL-17 higher levels of IFN-γ and Foxp3+	[66]

## Data Availability

Not applicable.

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
