# Peer review of "How to Improve Health with Biological Agents—Narrative Review"

_nutrients, 2022, doi:10.3390/nu14091700_

Round 1

Reviewer 1 Report

The article is well written, pleasant to read, with a clear and defined structure. However, when proposing a review article, it is important to choose the type of review. In this case and from my point of view, a narrative review has been chosen for the following reasons:

  • It gives a historical and general perspective of probiotics, prebiotics, synbiotics, postbiotics, paraprobiotics and psychobiotics.
  • It does not use any methodology (systematic review, scoping review, etc) for the elicitation of articles on which the conclusions are based.
  • Does not present a construction and writing following the scientific method and, therefore, may have biases in the writing and the potential risk of misleading the reader.
  • No limitations were detected in this review.

Therefore, my consideration is to add it to the title. Narrative review to differentiate from systematic reviews.

Author Response

The article is well written, pleasant to read, with a clear and defined structure. However, when proposing a review article, it is important to choose the type of review. In this case and from my point of view, a narrative review has been chosen for the following reasons:

  • It gives a historical and general perspective of probiotics, prebiotics, synbiotics, postbiotics, paraprobiotics and psychobiotics.
  • It does not use any methodology (systematic review, scoping review, etc) for the elicitation of articles on which the conclusions are based.
  • Does not present a construction and writing following the scientific method and, therefore, may have biases in the writing and the potential risk of misleading the reader.
  • No limitations were detected in this review.

Therefore, my consideration is to add it to the title. Narrative review to differentiate from systematic reviews.

Our answer: The title was changed into:

“How to improve health with biological agents - narrative review”

Reviewer 2 Report

This generally thoroughly written manuscript pays sufficient attention to recent work that has been carried out  in the field of probiotics around the globe. It is certainly publishable, but I'd like to make the following minor criticisms:

  1. Apart from the terms postbuiotics and parabiotics, a sufficiently widely used term to be mentioned in the review is metabiotic, the term preferred by one of the pioneers in this field, Prof. Boris Shenderov. A relevant reference is as follows: A. V. Oleskin and B. A. Shenderov, MICROBIAL COMMUNICATION AND MICROBIOTA-HOST INTERACTIONS: BIOMEDICAL, BIOTECHNOLOGICAL, AND BIOPOLITICAL IMPLICATIONS, Hauppauge (New York), Nova Science Publ., 2020  
  2. The long and thoroughly prepared list of probiotics in the relevant table in the manuscript lacks algal species, e.g., Chlorella, Scenedesmus, or Arthrospira (a cyanobacterium), that also meet the official criteria of probiotics.
  3. While discussing the impact of useful microorganisms on the nervous system, the authors should also give attention to microbial species that produce their own neurochemicals, e.g., catecholamines, serotonin, or histamine. I suggest referencing a couple of recent papers, e.g., A. V. Oleskin, B. A. Shenderov, and V. S. Rogovsky, Role of neurochemicals in the interaction between the microbiota and the immune and the nervous system of the host organism, Probiotics and Antimicrobial Proteins, 9 (2017), pp. 215–234.   

Author Response

This generally thoroughly written manuscript pays sufficient attention to recent work that has been carried out  in the field of probiotics around the globe. It is certainly publishable, but I'd like to make the following minor criticisms:

1. Apart from the terms postbuiotics and parabiotics, a sufficiently widely used term to be mentioned in the review is metabiotic, the term preferred by one of the pioneers in this field, Prof. Boris Shenderov. A relevant reference is as follows: A. V. Oleskin and B. A. Shenderov, MICROBIAL COMMUNICATION AND MICROBIOTA-HOST INTERACTIONS: BIOMEDICAL, BIOTECHNOLOGICAL, AND BIOPOLITICAL IMPLICATIONS, Hauppauge (New York), Nova Science Publ., 2020  

Our answer: The following information has been added:

In addition to the terms postbiotics and paraprobiotics, some works also include the term metabiotics, which refers to low molecular weight compounds that are metabolites, signaling molecules, or fragments of dead microbial cells. Metabiotics can be used as food additives, they can also influence the composition and functions of the gut microbiota. In addition, these compounds can also influence biochemical and behavioral responses as well as intracellular and intercellular information exchange [112,113]. Examples of compounds belonging to metabiotics are: bacteriocins, short-chain fatty acids, proteins, peptides, polysaccharides, vitamins or nucleic acids [113].

2. The long and thoroughly prepared list of probiotics in the relevant table in the manuscript lacks algal species, e.g., Chlorella, Scenedesmus, or Arthrospira (a cyanobacterium), that also meet the official criteria of probiotics.

Our answer: The following information has been added:

Some studies have reported that live microalgae (eg. Chlorella sp., Arthrospira sp., Schizochytrium sp.) can also be used as probiotics, especially in marine cultures, where they are expected to improve health and survival of marine animals [16]. Despite the proven positive impact on human and animal health of compounds extracted from algae (prebiotic effect), the evidence for probiotic benefits is still insufficient [16,17].

3. While discussing the impact of useful microorganisms on the nervous system, the authors should also give attention to microbial species that produce their own neurochemicals, e.g., catecholamines, serotonin, or histamine. I suggest referencing a couple of recent papers, e.g., A. V. Oleskin, B. A. Shenderov, and V. S. Rogovsky, Role of neurochemicals in the interaction between the microbiota and the immune and the nervous system of the host organism, Probiotics and Antimicrobial Proteins, 9 (2017), pp. 215–234.   

Our answer: The following information has been added:

In addition, some probiotic strains produce their own neurochemicals [12,116]. The strains of L. helveticus and L. delbrueckii subsp. bugaricus can produce norepinephrine and/or dopamine, moreover L. brevis, Lact. lactis and L. rhamnosus GG synthesize GABA, in turn serotonine was detected in B. subtilis and L. helveticus. Histamine, in turn, can be produced by numerous strains of both Gram-negative and Gram-positive bacteria, including by strains of L. acidophilus and Bacillus spp. [116]. On the other hand, E. coli, B. cereus and Lactobacillus spp. can produce catecholamines and their precursor - 2,3-dihydrophenylalanine (DOPA), which is converted to dopamine and norepinephrine [12].  

The following references have been added:

Camacho, F.; Macedo, A.; Malcata, F. Potential industrial applications and commercialization of microalgae in the functional food and feed industries: a short review. Mar. Drugs 2019, 17(6):312. DOI: 10.3390/md17060312

Perković, L.; Djedović, E.; Vujović, T.; Baković, M.; Paradžik, T.; Čož-Rakovac, R. Biotechnological enhancement of probiotics through co-cultivation with algae: future or a trend? Mar. Drugs 2022, 20(2):142. DOI: 10.3390/md20020142

Shenderov, B. A. Metabiotics: novel idea or natural development of probiotic conception. Microb. Ecol. Health Dis. 2013, 24, 10.3402/mehd.v24i0.20399. DOI:10.3402/mehd.v24i0.20399

Pihurov, M.; Păcularu-Burada, B.; CotârleÅ£, M.; Vasile, M. A.; Bahrim, G. E. Novel insights for metabiotics production by using artisanal probiotic cultures. Microorganisms 2021, 9(11), 2184. DOI:10.3390/microorganisms9112184

Oleskin, A.V.; Shenderov, B.A.; Rogovsky, V.S. Role of neurochemicals in the interaction between the microbiota and the immune and the nervous system of the host organism. Probiotics Antimicrob. Proteins 2017, 9(3):215-234. DOI: 10.1007/s12602-017-9262-1
